# Revisiting Adversarial Examples from the Perspective of Asymptotic Equipartition Property

## Abstract

Adversarial examples, which can mislead neural networks through subtle perturbations, continue to challenge our understanding, raising more questions than answers. This paper presents a novel perspective on interpreting adversarial examples through the Asymptotic Equipartition Property (AEP). Our theoretical analysis examines the noise within these examples, revealing that while normal noise aligns with AEP, adversarial noise does not. This insight allows us to classify samples in high-dimensional space as belonging to either the typical or non-typical set, corresponding to normal and adversarial examples, respectively. Our analyses and experiments show adversarial examples arise from AEP in high-dimensional space and derive some key properties regarding their quantity, probability, and information capacity. These findings enhance our understanding of adversarial examples and clarify their counterintuitive phenomena, such as adversarial transferability, the trade-off between robustness and accuracy, and robust overfitting.

## 1 Introduction

**Adversarial Examples**, small human-imperceptible perturbations of a benign input, which change the output of deep neural networks (DNNs), threaten various AI tasks, including traditional deep learning tasks Szegedy et al. (2014); Goodfellow et al. (2015); Carlini & Wagner (2018); Li et al. (2020), as well as popular LLM-based tasks Zhang et al. (2022); Zou et al. (2023); Liang et al.; Wu et al. (2024); Zhao et al. (2024). Although there have been a large amount of studies on adversarial examples Goodfellow et al. (2015); Ilyas et al. (2019), and several defense strategies proposed Metzen et al. (2017); Madry et al. (2018); Zhang et al. (2019); Kuang et al. (2023); Schlarmann et al. (2024); Zeng et al. (2024), the reason behind the susceptibility of adversarial examples remains an open question.

Previous works in this field have explained adversarial examples from various perspectives. Szegedy et al. (2014) considered adversarial examples as low-probability, high-dimensional pockets in the manifold. Goodfellow et al. (2015) viewed them as fluctuations resulting from the linear behavior in the high-dimensional nature of the input space. Gilmer et al. (2018) hypothesized that this behavior arises from the high-dimensional geometry of data manifolds and low but non-zero error rates. More broadly, Ilyas et al. (2019) argued that adversarial examples are features rather than bugs, suggesting that the features learned by DNNs can be divided into robust and non-robust features, and that adversarial vulnerability is a fundamental consequence of the dominant supervised learning paradigm. Tsipras et al. (2019) showed that representations learned by standard and robust models are fundamentally different, sparking debates on whether there exists a trade-off between adversarial robustness and clean accuracy. Zhang et al. (2019) proposed TRADES, which characterizes this trade-off theoretically, algorithmically, and experimentally. Conversely, Raghunathan et al. (2020) argued that infinite data can eliminate this trade-off. Furthermore, Yang et al. (2020) proved that the trade-off in deep learning is not inherent but a consequence of current methods for training robust networks.

Except for the trade-off problem, adversarial examples raise many other counterintuitive behaviors. One intriguing behavior is adversarial transferability: the phenomenon where adversarial perturbations computed for one model can transfer to other independently trained models Papernot et al. (2016a); Cheng et al. (2019). Robust adversarial training also exhibits overfitting, termed robust overfitting Rice et al. (2020), where robust accuracy rises immediately after the first learning rate decay and

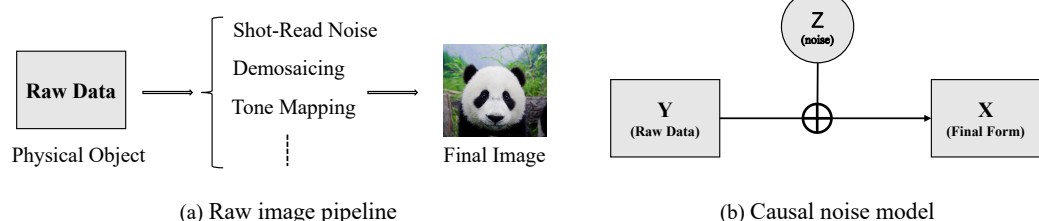

(a) Raw image pipeline          (b) Causal noise model

Figure 1: (a) is raw image pipeline. The camera sensor captures the raw data, then optical processing is required to transform its noisy linear intensities into the final image. (b) is causal graphs with $Y$, $Z$ causing $X$, where $Y$ is the raw data (the real-world physical object), $Z$ is the perturbation introduced during the entire imaging process, $X$ is the final image. The causal process can correspond to the raw image pipeline.

then decreases beyond this point. Additionally, adversarial learning requires a high-capacity network and more training data Madry et al. (2018); Schmidt et al. (2018) than standard learning. Despite abundant theories and empirical experiments, it is still not fully understood why adversarial examples lead to such behaviors across the various aspects mentioned above.

To further explore this inquiry, we focus on image data and establish a causal noise model to simulate the image generation process, as illustrated in Figure 1. We assume the existence of an underlying noise-free dataset $Y$, with any variability attributed to additional noise $Z$. We hypothesize that the abnormal behavior of samples is driven by this noise $Z$. As discussed in the subsequent section, we find that normal noise adheres to the Asymptotic Equipartition Property (AEP) Shannon (1948), whereas adversarial noise does not. AEP, a fundamental property of samples drawn from a probability distribution, arises from the weak law of large numbers. According to AEP theory, samples in high-dimensional space can be divided into a typical set and an non-typical set. The behavior of samples is largely governed by the typical set, which contains those that satisfy the AEP criteria, while adversarial samples predominantly fall within the non-typical set. This hypothesis is empirically validated by training Deep Neural Networks (DNNs) on artificially generated datasets containing both sets and assessing their vulnerability to adversarial attacks. In essence, adversarial examples can be understood as a manifestation of the AEP in high-dimensional space.

Leveraging the Asymptotic Equipartition Property (AEP), we identify several key characteristics of adversarial examples that help explain their counterintuitive phenomena, including adversarial transferability, the trade-off between robustness and accuracy, and robust overfitting:

- High-dimensional data can be divided into typical and non-typical sets. Normal samples correspond to the typical set, while adversarial samples belong to the non-typical set. In essence, adversarial examples can be understood as a manifestation of the AEP in high-dimensional space.

- Adversarial vulnerabilities occur because deep neural networks are unable to learn the intrinsic features of non-typical sets in high-dimensional space. This limitation stems from the fact that the data samples used in standard training conform to the AEP and belong to the typical set. As a result, the model is not exposed to or capable of learning the features of non-typical sets.

- In high-dimensional spaces, adversarial examples belong to the low-probability set, while normal examples reside in the high-probability set. Interestingly, the number of adversarial examples significantly exceeds that of normal examples. As a result, robust learning necessitates larger models and more extensive datasets to effectively capture both typical and non-typical patterns.

## 2 ASYMPTOTIC EQUIPARTITION PROPERTY

In information theory, the Asymptotic Equipartition Property (AEP) Shannon (1948) is a general property of the output samples from a probability distribution. It is fundamental to the concept of the typical set used in theories of data compression and is a direct consequence of the weak law of large numbers. The following Theorem 1 formalizes the classical AEP [1].

---

[1]For more details of the AEP, we refer the reader to Shannon (1948); Algoet & Cover (1988); Cover (1999).

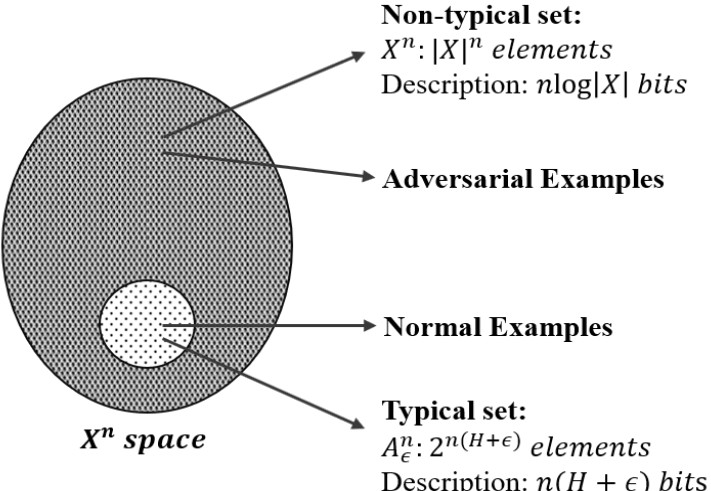

Figure 2: Visualization of the AEP from our perspective. We divide the samples into two parts, *i.e.,* the typical set and the non-typical set, which correspond to the normal and adversarial sample, respectively. According to the AEP theory, the number of samples in the typical set is $2^{n(\mathcal{H}+\epsilon)}$ that is smaller than that of the non-typical set $|\mathcal{X}|^n$. In addition, we give the definition of maximum coding descriptions for the typical set and non-typical set, where the bits of typical set is $n(\mathcal{H}+\epsilon)$, while that of non-typical set is $n\log|\mathcal{X}|$. This can explain why a high-capacity network is required for adversarial training, which is related to our conclusion in the Section 4.5.

**Theorem 1.** (AEP): if $x_1, x_2, \ldots$ are i.i.d. $\sim p(x)$, then

$$-\frac{1}{n}\log p(x_1, x_2, \ldots, x_n) \to \mathcal{H}(X), \tag{1}$$

where $H(X)$ denotes the entropy rate of $X$.

**A Toy Example.** Let us define the random variable $x \in \{0, 1\}$ has a probability mass function, where $p(1) = p$ and $p(0) = q$. If $x_1, x_2, \ldots, x_n$ are i.i.d. random samples taken from $P(x)$, the probability of a sequence $p(x_1, x_2, \ldots, x_n)$ is $\prod_{i=1}^n p(x_i)$. If there are two sequences, *i.e.,* $(1, 0, 1, 1, 0, 1)$ and $(0, 0, 0, 0, 0, 0)$, and $p(1) = p = 0.8$, we can obtain the following:

$$\begin{aligned}
p(1, 0, 1, 1, 0, 1) &= p^4 q^2 = 0.0164, \\
p(0, 0, 0, 0, 0, 0) &= p^0 q^6 = 0.0000064.
\end{aligned} \tag{2}$$

It is clear that not all sequences of the same length have the same probability. Assuming $n \to \infty$, the number of 1's in the sequence is close to $np$ with high probability, and all such sequences have the same probability $2^{-n\mathcal{H}}$. The AEP indicates that samples meeting the property of AEP belong to a high-probability set and determine the overall behavior of all samples.

That is, the AEP states that $-\frac{1}{n}\log p(x_1, x_2, \ldots, x_n)$ is close to the entropy $\mathcal{H}$, where $x_1, x_2, \ldots, x_n$ are the i.i.d. random variables and $p(x_1, x_2, \ldots, x_n)$ is the probability of observing the sequence $(x_1, x_2, \ldots, x_n)$. Thus, the probability $p(x_1, x_2, \ldots, x_n)$ assigned to an observed sequence will be close to $2^{-n\mathcal{H}}$.

According to Cover (1999), AEP theory allows us to divide any high-dimensional dataset into two independent sets: the typical set (i.e., the entropy of the samples is close to the true entropy) and the non-typical set (i.e., samples outside the typical set), as shown in Figure 2. Therefore, the definition of the typical set is as follows:

**Typical Set.** The typical set $\mathcal{A}_\epsilon^{(n)}$ *w.r.t.* $p(x)$ is the set of sequences $(x_1, x_2, \ldots, x_n) \in \mathcal{X}^n$ with the following property:

$$2^{-n(\mathcal{H}+\epsilon)} \le p(x_1, x_2, \ldots, x_n) \le 2^{-n(\mathcal{H}-\epsilon)}, \tag{3}$$

where $\epsilon$ is a constant.

We introduce some important properties of the typical set $\mathcal{A}_\epsilon^{(n)}$ as follows, which serve as the fundamental preliminaries of this paper.[2]

**Properties.** If $(x_1, x_2, \ldots, x_n) \in \mathcal{A}_\epsilon^{(n)}$, we have:

(1). $\mathcal{H}(X) - \epsilon \leq -\frac{1}{n} \log p(x_1, x_2, \ldots, x_n) \leq \mathcal{H}(X) + \epsilon$, which is determined by the definition of the typical set.

(2). $Pr(\mathcal{A}_\epsilon^{(n)}) > 1 - \epsilon$, for any small number $\epsilon$ together with sufficiently large $n$.

(3). The scale of the typical set $|\mathcal{A}_\epsilon^{(n)}| \leq 2^{n(\mathcal{H}(x)+\epsilon)}$.

(4). $(1 - \epsilon)2^{n(\mathcal{H}(x)-\epsilon)} \leq |\mathcal{A}_\epsilon^{(n)}|$ for a sufficiently large $n$.

# 3 ANALYSIS AND VERIFICATION

## 3.1 CAUSAL NOISE MODEL

Modern digital cameras strive to render a pleasant and accurate image of the real world, simulating what the human eye sees Szeliski (2010). However, the raw sensor data from a camera does not resemble a photograph, requiring many processing stages to transform its noisy linear intensities into their final form. These stages include shot and read noise Hasinoff (2014), demosaicing Gharbi et al. (2016), and tone mapping Debevec & Malik (2008), as shown in Figure 1(a). Each of these steps may influence the final observed data.

For simplicity, we model these processes as a noise graph model, visualized as an exemplar probability graph in Figure 1(b). We assume that $Y$ represents the raw data (i.e., the real-world physical object), which is pure and unpolluted. We define $Z$ as the noise introduced during the overall imaging process, with any uncertainties arising from this additional noise $Z$. $X$ as the final image, where the appearance of $X$ is influenced by both the object $Y$ and the noise $Z$. From the noise graph model, we can define valid perturbations of data through the lens of causality. Generating an adversarial example is equivalent to perturbing the factors that produce $X$ in the graph model, where we posit that an adversarial perturbation is an intervention on $Z$. We exclude intervention on $Y$ because it would alter the actual objects in the image, which contradicts the setting of human-imperceptible perturbations. Therefore, we focus on the influence of the noise $Z$ on the final image. Generally, a DNN takes $X$ as input and directly outputs the prediction $Y$, which can be formulated as $p(Y|X) = \frac{p(Y)p(X|Y)}{p(X)}$.

Experiments show that deep neural networks are not sensitive to small and normal noise, such as Gaussian or uniform noise. Adding such noise to data samples typically does not change the model's output. However, adversarial noise can mislead the network into producing incorrect results. We assumes all noise is relatively small (e.g., image noise with a magnitude of $8.0/255$), remaining imperceptible to the human eye. The differing effects of normal and adversarial noise demonstrate that, despite their similar appearance, they possess fundamentally different properties. Numerous studies have attempted to train classifiers to distinguish between adversarial and normal samples Metzen et al. (2017); Cohen et al. (2020). However, the precise nature of this fundamental difference remains unknown

## 3.2 DISENTANGLING NORMAL AND ADVERSARIAL EXAMPLES

To differentiate between normal and adversarial noise, we associate the AEP of data with noise $Z$. Through AEP, samples in high-dimensional space are divided into typical and non-typical sets. We prove that normal samples and adversarial samples correspond to typical and non-typical sets, respectively.

---

[2]The corresponding proofs are provided in the supplementary material and are useful for understanding the adversarial examples.

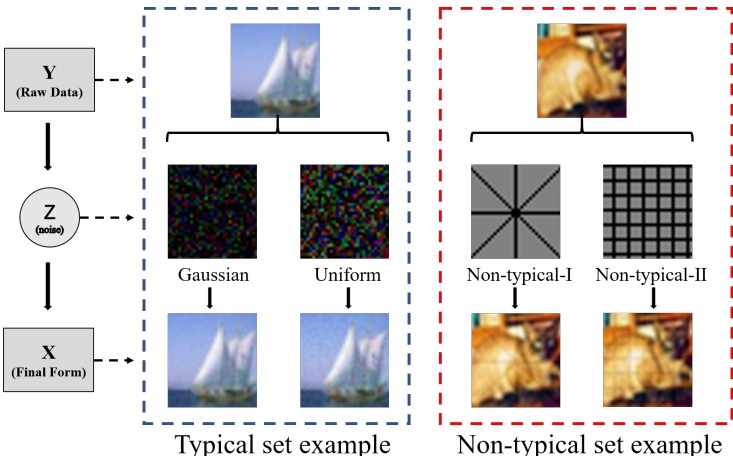

Figure 3: We have constructed typical and non-typical datasets on CIFAR-10. For typical set examples, we add AEP-compliant noise to clean examples, such as Gaussian noise or uniform noise. For non-typical set examples, we artificially construct two types of noise that do not meet AEP. Finally, the final image is synthesized through our causal noise model.

Let $Z = (z_1, z_2, \ldots, z_n)$, where $z_1, z_2, \ldots, z_n$ are i.i.d. samples from $P(Z)$, and $n = C \times W \times H$. For example, in CIFAR-10 Krizhevsky et al. (2009) data, $Z \in \mathbb{R}^n$ and $n = 3 \times 32 \times 32$. We have the following Lemmas:

**Lemma 1.** Normal samples $X_{norl}$ belong to the typical set.

Throughout the image generation process, various factors may influence the final image. Normally, this noise adheres to the AEP. For instance, shutter noise follows a Poisson random variable, and read noise approximates a Gaussian random variable with zero mean and fixed variance. Therefore, generally, high-dimensional noise $Z$ can be considered as independent random variables following distribution $p(Z)$, such as Gaussian, Poisson, or exponential distribution. When sampling normal noise from $p(Z)$, it will conform to the characteristics of AEP, making $Z$ the noise in the typical set. Consequently, the sampled image becomes a normal sample $X_{norl}$, which belongs to the typical set.

**Lemma 2.** Adversarial samples $X_{adv}$ belong to the non-typical set.

The majority of existing methods for generating adversarial perturbations, such as FGSM or PGD, rely on model gradients. Black-box attacks similarly utilize gradient estimation to create adversarial samples. Their formulations can be simplified as follows:

$$Z_k = \Pi\Big(Z_{k-1} + \alpha \cdot \text{sign}(\nabla_x \mathcal{L}(F_\theta(X + Z_{k-1}), Y))\Big),$$

where $F$ represents the neural network model with weights $\theta$, $\mathcal{L}$ denotes the cross-entropy loss function, $\Pi$ stands for the projection function, $\alpha$ indicates the step size, and $Z_k$ signifies the adversarial perturbation at step $k$.

In this scenario, due to the intervention of adversarial noise, the true distribution of $Z$ becomes indeterminate, making it difficult to ascertain whether $Z$ conforms to AEP. To tackle this challenge, we adopt a causal perspective and hypothesize that the adversarial noise $Z_{adv}$ is drawn from the distribution $P(Z|G)$, where $G$ serves as the prior for generating noise (based on gradient information). Consequently, if $Z_{adv}$, sampled from $P(Z|G)$, does not adhere to the AEP, it can be classified as belonging to the non-typical set. As a result, the adversarial samples $X_{adv}$ also belong to this non-typical set. A more detailed proof is available in Appendix D.

Drawing from **Lemma 1** and **Lemma 2**, it becomes clear that, although the human eye may not detect subtle differences between normal noise and adversarial noise, significant mathematical and statistical distinctions exist, driven by the Asymptotic Equipartition Property (AEP) of the data. According to the properties of AEP, high-dimensional data can be divided into two categories: typical and

Table 1: Generalizability attack of the typical noise and non-typical noise across different datasets and networks. NT denote non-typical. The perturbation budget of $\epsilon = 8/255$.

| Datasets and Networks | Clean | Gaussian | Uniform | NT-I | NT-II |
|:---:|:---:|:---:|:---:|:---:|:---:|
| CIFAR-10 | 91.9 | 91.3 (−0.6) | 89.7 (−2.2) | 76.5 (−15.4) | **72.1 (−19.8)** |
| SVHN | 95.7 | 95.7 (−0.0) | 95.4 (−0.3) | 91.1 (−4.60) | **87.1 (−8.60)** |
| TinyImage | 51.8 | 51.5 (−0.3) | 51.3 (−0.5) | 47.6 (−4.20) | **39.2 (−12.6)** |
| ResNet | 91.9 | 91.3 (−0.6) | 89.7 (−2.2) | 76.5 (−15.4) | **72.1 (−19.8)** |
| VGG | 91.1 | 91.0 (−0.1) | 90.2 (−0.9) | 88.8 (−2.30) | **83.9 (−7.20)** |
| DenseNet | 92.4 | 90.9 (−1.5) | 88.4 (−4.0) | 82.8 (−9.60) | **74.2 (−18.2)** |
| MobileNet | 90.1 | 88.4 (−1.7) | 84.7 (−5.4) | 79.1 (−11.0) | **71.4 (−18.7)** |

non-typical sets. Normal samples fall within the typical set, while adversarial samples are classified as belonging to the non-typical set.

### 3.3 CONSTRUCTING TYPICAL AND NON-TYPICAL EXAMPLES

Our proposed approach is based on the premise that both typical and non-typical sets exist in high-dimensional space under the AEP. To investigate this, we aim to construct artificial typical and non-typical sets, and then train deep neural networks (DNNs) on these datasets to analyze their properties. Assuming that $Y$ consists entirely of clean data, our focus shifts to the characteristics of the noise $Z$. Specifically, when $Z$ represents typical noise, $X$ is classified as a typical sample; conversely, when $Z$ represents non-typical noise, $X$ is classified as an non-typical sample.

To construct the typical set initially, we introduce AEP-compliant noise into the clean examples $Y$. This noise can be randomly sampled from common distributions like Gaussian or uniform distributions. Conversely, for the non-typical set, we introduce noise that deviates from the AEP when applied to the clean examples $Y$. Indeed, generating noise that doesn't adhere to the AEP is relatively straightforward due to the abundance of non-typical noise types. There are two straightforward methods to create samples for the non-typical set. One involves leveraging information from trained DNNs, where non-typical noise is generated using the DNN gradient as a prior. The other method entails generating non-typical noise relevant to the sample space, akin to a form of universal adversarial perturbation Moosavi-Dezfooli et al. (2017); Liu et al. (2019). Here, we concentrate solely on the latter approach, which can be practically crafted through simple manual disturbances, as depicted in Figure 3. The noise labelled as non-typical-I and non-typical-II is custom-designed by us and does not conform to the AEP.

To confirm the efficacy of the non-typical noise we generated for adversarial attacks, we perform experiments across various datasets, comparing its impact with that of typical noise. The results are detailed in Table 1, wherein we assess model performance on CIFAR-10, SVHN, and TinyImageNet datasets. Notably, employing typical noise as an adversarial perturbation results in minimal accuracy loss for DNNs, whereas the utilization of non-typical noise leads to a notable decrease in accuracy. This observation underscores the general characteristic of non-typical noise, indicating its resilience across different datasets. Subsequently, we assess performance across various backbone architectures such as ResNet, VGG, DenseNet, and MobileNet. Table 1 further illustrates that non-typical noise markedly reduces model accuracy. This experiment elucidates the transferability of adversarial examples and underscores the presence of universal adversarial perturbation.

Moreover, we assess performance under robust adversarial training, which differs from standard adversarial training Madry et al. (2018). During training, we initially employ the PGD attack to generate adversarial examples and then introduce artificially constructed noise, as previously described. Consequently, we adapt the original adversarial examples and utilize either typical or non-typical adversarial examples for training. During testing, we similarly introduce corresponding noise to input samples. All models are evaluated using a 10-step PGD attack. We term this tailored adversarial training as AEP-based adversarial training (AEP AT), as shown in Figure 4. The experimental results are presented in Figure 5(a). Notably, the model trained on adversarial examples with typical noise performs well on clean examples with a certain degree of robustness. Similar to standard adversarial training, the robustness accuracy is lower than the clean accuracy, albeit consistent with standard

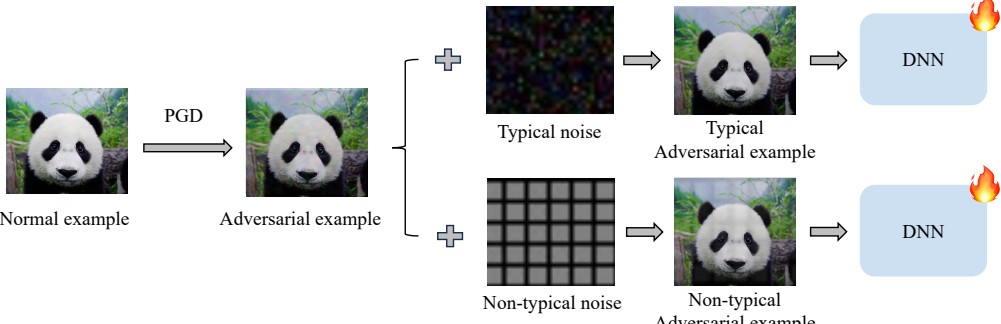

Figure 4: The pipeline of AEP-AT.

practices. In contrast, for the model trained on adversarial examples with two types of non-typical noise, we observe that clean accuracy is lower than robustness accuracy. This discrepancy indicates that the non-typical noise we introduced improves the model's ability to fit adversarial examples, which contrasts with the results from training on typical noise. Therefore, we argue that the data space is divided into two distinct domains: one consisting of typical samples and normal examples, and the other comprising non-typical samples and adversarial examples.

Based on the preceding experiment, we deduce that typical samples and normal examples share similar properties, while non-typical samples and adversarial examples exhibit analogous characteristics. Consequently, we contend that the typical set aligns with normal examples, whereas the non-typical set corresponds to adversarial examples. Building upon the insights from Section **??**, we demonstrate that the typical set and adversarial examples are interchangeable.

## 4 EXPLAINING COUNTERINTUITIVE BEHAVIORS

From our new perspective, our theory and experiments not only give a clear insight into adversarial examples, but also explain some counterintuitive behaviors, such as standard training not robust, the trade-off between robustness and accuracy, adversarial transferability, and robust overfitting, *etc*.

### 4.1 STANDARD TRAINING IS NOT ROBUST

Ilyas et al. (2019) argued when training on the standard dataset, non-robust features take on a large role in the resulting learned DNNs. From our perspective, we argue that adversarial vulnerability is due to the DNNs not fitting the features from the non-typical in the high-dimensional space. The fundamental reason is that there are no non-typical samples in the standard training dataset, so the DNNs have no chance to learn the non-typical features. To verify this point, we suppose that the typical set is the smallest high-probability set.

From the properties of the typical set, when $n$ is sufficiently large, the probability of the typical samples (normal examples) have $Pr(\mathcal{A}_\epsilon^{(n)}) > 1 - \epsilon$, where $\epsilon$ is any small number. In turn, we get a probability of 0 for the non-typical samples (adversarial examples). However, one interesting thing is that, in the entire $n$-dimensional space, the number of samples in the non-typical set is far more than that in the typical set. Specifically, the number of samples in the typical set is about $2^{n(\mathcal{H}\pm\epsilon)}$, and the number of samples in the entire space is $|\mathcal{X}|^n$, where $|\mathcal{X}|$ is the size of the state-space, we have

$$\lim_{n \to \inf} \frac{2^{n(\mathcal{H}\pm\epsilon)}}{|\mathcal{X}|^n} = 0. \tag{4}$$

Thus, $\mathcal{A}_\epsilon^{(n)}$ is a fairly small set that contains most of the probability. Now we demonstrate that the typical set has the same number of samples as the smallest set.

**Definition:** For each $n = 1, 2, \ldots$, let $\mathcal{B}_\delta^{(n)} \in \mathcal{X}^n$ be any set with

$$Pr(\mathcal{B}_\delta^{(n)}) > 1 - \delta. \tag{5}$$

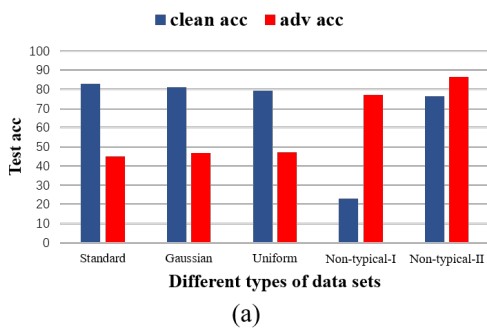 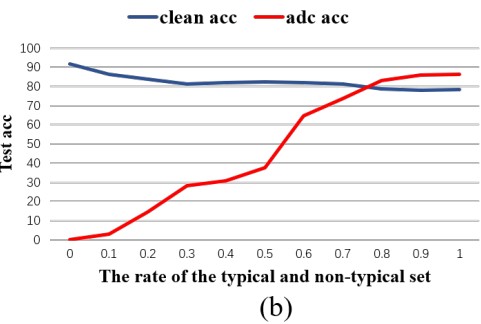

(a)                                                                (b)

Figure 5: **Left (a):** Under the adversarial training setting, clean accuracy and robust accuracy for models trained with the typical and non-typical datasets. **Right (b):** Trade-off between clean accuracy and robust accuracy. We adjust the ratio of the typical samples and the non-typical samples to achieve a trade-off.

We assume that $\mathcal{B}_{\delta}^{(n)}$ must have a significant intersection with $\mathcal{A}_{\epsilon}^{(n)}$ and therefore must have about as many samples.

**Theorem 2.** In Cover (1999), assume $x_1, x_2, \ldots$ i.i.d. $\sim p(x)$, for $\delta < \frac{1}{2}$ and any $\delta' > 0$, if $Pr(\mathcal{B}_{\delta}^{(n)}) > 1 - \delta$, when $n$ is sufficiently large, then we have

$$\frac{1}{n} \log |\mathcal{B}_{\delta}^{(n)}| > \mathcal{H} - \delta'. \tag{6}$$

Thus, $\mathcal{B}_{\delta}^{(n)}$ must have at least $2^{n\mathcal{H}}$ sample, to first-order in the exponent, while $\mathcal{A}_{\epsilon}^{(n)}$ has $2^{n(\mathcal{H}\pm\epsilon)}$. Therefore, $\mathcal{A}_{\epsilon}^{(n)}$ is about the same size as the smallest high probability set. This interesting result shows that the number of samples in the typical set is extremely small compared to the total number of samples in the entire space, but they do exist and appears with a high probability.

From the above theories, we conclude that, in general, the datasets we collect are from the typical set, and our DNNs work on the typical set, regardless of training or testing, so the DNNs can have good generalization. However, when the DNNs face adversarial examples (non-typical set), which they have not learnt, they are deceived.

### 4.2 TRADE-OFF BETWEEN ROBUSTNESS AND ACCURACY

In the realm of robust adversarial training, there has been considerable debate regarding the existence of a trade-off between robustness and accuracy. A prevailing notion suggests that robustness and accuracy are mutually detrimental Zhang et al. (2019); Tsipras et al. (2019). Nonetheless, some studies have contended that certain benchmark datasets exhibit class separation Yang et al. (2020), positing that robustness can be upheld while enhancing accuracy with an infinite dataset Raghunathan et al. (2020).

From our novel perspective, we argue that a delicate balance exists between robustness and accuracy in the current learning paradigm. This trade-off arises from the partitioning of high-dimensional space into typical and non-typical sets, each characterized by distinct properties. While, in theory, infinite training data and a network with suffi-

Table 2: Robustness and accuracy comparison of AEP-AT with Standard AT on different datasets

| Datasets | Standard AT | | AEP AT | |
|---|---|---|---|---|
| | Clean | Adv acc | Clean | Adv acc |
| CIFAR-10 | 85.7 | 48.3 | 78.4 | 86.5 |
| SVHN | 93.6 | 51.2 | 92.1 | 93.2 |
| TinyImage | 46.8 | 21.1 | 45.6 | 49.6 |

cient capacity could accommodate all possible samples, in practice, the typical set represents a high-probability domain, while the non-typical set contains a disproportionately larger number of samples. Due to limitations in the capacity of current networks, achieving both high robustness and

clean accuracy simultaneously is difficult. Therefore, we aim to navigate this trade-off between robustness and accuracy within these constraints.

To validate this assertion, we implement AEP-based adversarial training (AEP AT). Here, we augment the adversarial examples generated by the PGD attack with the non-typical pattern (e.g., non-typical-II), then utilize these modified adversarial examples for training. AEP AT is evaluated across different datasets using a 10-step PGD attack, and the outcomes are detailed in Table 2. Relative to standard adversarial training, AEP-AT maintains a higher robustness accuracy compared to clean accuracy. This suggests the presence of a balance point where robustness and accuracy stabilize, rather than exhibiting bias toward either extreme. To pinpoint this equilibrium, we train the DNN using both clean samples and modified adversarial samples, varying the ratio of the two and monitoring the resulting model's robustness and accuracy. The findings, depicted in Figure 5(b), demonstrate that our trained model attains an optimal trade-off state between robustness and accuracy. This outcome aligns precisely with our expectations, affirming our hypothesis.

### 4.3 Robustness Overfitting

Rice et al. (2020) highlighted the presence of robust overfitting in robust adversarial training, where robust accuracy initially increases following the first learning rate decay but declines thereafter. Overfitting in machine learning typically arises due to either an insufficient size of training data or an inconsistency between the feature distributions of training and test data.

Viewed through the lens of AEP, the high-dimensional data is partitioned into two domains: a typical set and a non-typical set, each characterized by distinct feature distributions. Our research demonstrates that the non-typical set contains significantly more samples than the typical set. Furthermore, as current adversarial training employs Projected Gradient Descent (PGD) to generate adversarial examples for training, the adversarial noise is intricately linked to the input samples. Consequently, the model predominantly learns features specific to the non-typical set related to the training samples, impeding generalization to test samples and leading to robust overfitting.

### 4.4 Adversarial Transferability

Another crucial aspect of adversarial examples is their transferability, a phenomenon where perturbations crafted for one model can effectively target another, regardless of their training Papernot et al. (2016a); Cheng et al. (2019). Ilyas et al. (2019) posit that due to the likelihood of two models learning similar non-robust features, perturbations manipulating such features can affect both models. This perspective holds merit to some extent. As outlined in Section 4.1, standard benchmark datasets typically comprise samples from a common set. Consequently, deep neural networks (DNNs) are trained predominantly on these standard samples, learning analogous features. The distinction from Ilyas et al. (2019) lies in our assertion that these common set features arise from the high-dimensional characteristics of external noise, rather than inherent non-robust features within the samples. An adversary manipulates the AEP of pristine samples using the gradient information of DNNs, thus converting samples from the typical set into the non-typical set. It is important to note that adversaries utilize model information to craft adversarial examples. However, the AEP remains unaffected by the model's architecture or the dataset's category; it is solely linked to the high-dimensional data distribution. Hence, adversarial examples can transcend different model architectures, rendering them universal.

To validate this claim, we conducted several comparative experiments in Section 3.3, employing both typical and non-typical samples to evaluate various model architectures and benchmark datasets. The results, depicted in Table 1, support our assertion that the AEP bias in high-dimensional space underlies adversarial examples, independent of model architecture and datasets.

### 4.5 Bigger Model and More Data

Many works have found that adversarial training not only consumes computational resources but also requires a high-capacity network and more training data to improve the robustness of the model Madry et al. (2018). Now, from the perspective of AEP-based data compression, we try to explain why a larger model and more data are needed to improve robustness. We design a coding scheme for samples in high-dimensional space. The size of the typical set does not exceed $2^{n(\mathcal{H}+\epsilon)}$, so the

index of all these samples can be encoded by no more than $n(\mathcal{H} + \epsilon)$ bits. Similarly, the size of the non-typical set is about $|\mathcal{X}|^n$, so we can encode the index of each sample in the non-typical set by using no more than $n \log |\mathcal{X}|$ bits. A model with limited capacity is usually only trained on the typical set, so it only needs to accommodate the information with $n(\mathcal{H} + \epsilon)$ bits. Under adversarial training, the model must fit not only the typical samples but also the non-typical samples. However, the non-typical set information has $n \log |\mathcal{X}|$ bits, which is much larger than the $n(\mathcal{H} + \epsilon)$ bits of the typical set. Such analyses show the original model capacity is insufficient, and a high-capacity model is needed to better accommodate the increased information.

On the other hand, there are many works to improve the robustness of the DNNs by adding additional training data Schmidt et al. (2018). From our perspective, it is equivalent to increasing the training data of the non-typical samples (adversarial examples), which can be regarded as another form of adversarial training. In this way, the DNNs learn the features from the non-typical set and can better fit the non-typical set (adversarial examples). Therefore, additional data not only improves the robustness of the model but also can reduce overfitting.

## 5 CONCLUSIONS

In this paper, we revisit adversarial examples from a new perspective: asymptotic equipartition property (AEP). We decompose and construct normal and adversarial samples, further explore the consequences of AEP causing the model's adversarial vulnerability. We further derive important properties of normal and adversarial samples in terms of quantity, probability, and information capacity, thus providing explainable reasons for a series of related phenomena.

The goal of this work is to explore and explain the adversarial phenomenons. Our findings not only provide novel insights into adversarial examples but also serve as inspiration for researchers to devise new defense or attack algorithms. Importantly, within the current learning paradigm, complete immunity to adversarial attacks remains elusive. Hence, the pursuit of designing a new learning paradigm to align models more closely with human cognition represents a valuable research trajectory.

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

## A    RELATED WORK

### A.1    ADVERSARIAL ATTACK

Adversarial example was first proposed in Szegedy et al. (2014), following by a series of adversarial attacks to mislead DNN predictions by altering the inputs with human-imperceptible perturbation Moosavi-Dezfooli et al. (2016); Papernot et al. (2016b); Carlini & Wagner (2017). The Fast Gradient Sign Method (FGSM) Goodfellow et al. (2015) is a classical adversarial attack, for an input image, FGSM uses the gradient of the loss w.r.t. the input image to create an adversarial image. Another strong attack method is Project Gradient Descent (PGD) attack Madry et al. (2018), creates the adversarial examples by using a multi-step projected gradient descent, which is the most popular method to test adversarial robustness. Moosavi-Dezfooli et al. (2017); Liu et al. (2019) constructed a single adversarial noise, termed universal adversarial perturbation (UAP), is sufficient to fool most images from a data distribution with a given CNN model.

In addition, differing from aforementioned methods that require full knowledge of a DNN, black-box attacks are more practical, which uses the adversarial transferability of adversarial examples. Previous work Wu et al. (2018); Dong et al. (2019) shows that adversarial samples generated by one model can attack other models with a high probability, which grants the attacker more flexibility. Another type of black-box attack is a query-based attack Andriushchenko et al. (2019); Chen et al. (2020). Query-based attacks update the perturbation iteratively to optimize the attack objective.

From our perspective, all these attack algorithms are looking for non-typical set samples in the data sample space. Both adversarial transferability and UAP are based on the properties of non-typical set.

### A.2    ADVERSARIAL DENFENSE

With the rapid development of attack methods, considerable efforts have been devoted to defending against adversarial examples, such as defensive distillation Papernot et al. (2016c), manifold-projection Samangouei et al. (2018), pre-processing Guo et al. (2018); Yang et al. (2019), verification and provable defenses Raghunathan et al. (2018); Salman et al. (2019), and Adversarial Training Goodfellow et al. (2015); Madry et al. (2018); Cranko et al. (2019). AT augments the training procedure with adversarial examples produced by adversarial attacks, in details, the adversarial training is a kind of minimax optimization problems, which can be formulated as:

$$\min_{\theta} E\bigg[ \max_{x_{\text{adv}}} \mathcal{L}\big( F_{\theta}\big(x_{\text{adv}}\big), y \big) \bigg], \tag{7}$$

where $F_{\theta}$ is a DNN model with parameters $\theta$, and $\mathcal{L}$ is the loss function of the DNN. This objective has an adversarial form. The inner maximization conducts a typical adversarial attack. For a given image $x$, it aims to find an $x_{\text{adv}}$ within the $\epsilon$-ball of $x$, such that the training loss is maximized, *i.e.* the DNN is fooled. The inner maximization can be solved approximately, using PGD attack.

From our perspective, all current defense methods can be divided into two categories. One is based on adversarial training, which enables the model to learn non-typical set features, thereby making the model robust. The other is to transform non-typical set samples into typical set samples. so that the input samples conform to the features of the typical set learned by the model.

### A.3    ADVERSARIAL EXPLAINABILITY

Several works have been devoted to explaining the phenomenon of adversarial examples, such as boundary tilting Tanay & Griffin (2016), local linearity Goodfellow et al. (2015), and test error in noise Fawzi et al. (2016). However, the closest to our work is Ilyas et al. (2019). Ilyas et al. (2019) argued that adversarial examples are not bugs, but features. They explicitly disentangled robust and non-robust features in standard datasets. Compared to them, the concept of typical (non-typical) set that we have proposed is similar to that of non-robust (robust) feature, but the key differentiating aspect of our perspective is that we argue that adversarial examples are caused by the interference of external noise, rather than inherent features of the samples themselves. On the other hand, regarding the typical and non-typical set, we have strict mathematical definitions, not abstract descriptions.

## B  IMPLEMENTATION DETAILS.

In our work, we customized a special adversarial training, termed the AEP-based adversarial training (AEPAT). Specially, in the training phase, we first use the PGD attack to generate adversarial examples, where step size = 2/255 with the iteration of 7 and the perturbation budget of = 8. Then we add artificially constructed noise to them as generated before. Therefore, we modify the original adversarial examples and use the typical or non-typical adversarial examples for adversarial training. The initial learning rate $\lambda = 0.1$ and the learning rate schedule is [0.1, 0.01, 0.001], the decay epoch schedule is [70, 75]. The training scheduling of 80 epochs. We performed standard data augmentation including random crops and random horizontal flips during training. In the testing phase, we also add corresponding noise to the input samples. All models are evaluated with 10 steps PGD attack, where step size = 2/255 and perturbation budget = 8.

## C  LIMITATIONS

We explain the generation of adversarial examples and the reasons for adversarial vulnerability in commonly trained models from the perspective of AEP, offering a higher-dimensional interpretation. It derives important characteristics of non-robust representations in terms of quantity, probability, and information capacity, providing explanatory reasons for a range of related phenomena. These insights are not offered by other explanatory methods.

In the exploration of explanations based on AEP, adversarial examples can be generated in various ways, and different types of adversarial examples may have unique characteristics and properties. However, we only consider two types of non-typical noise, which may result in a dataset that is not sufficiently rich and comprehensive, thereby limiting the generalizability of the explanations.

Therefore, the interpretability of neural network models regarding adversarial examples still faces many challenges and limitations. Continued efforts in future research are needed to find new methods and strategies to overcome these challenges.

## D  PROOFS

**Theorem 1.** (AEP): if $x_1, x_2, \ldots$ are i.i.d. $\sim p(x)$, then

$$-\frac{1}{n} \log p(x_1, x_2, \ldots, x_n) \to \mathcal{H}(X), \tag{8}$$

where $H(X)$ denotes the entropy rate of $X$.

*Proof.* Function of independent random variables are also independent random variables, Thus, since the $x_i$ are i.i.d., so are $\log p(x_i)$. Hence by the weak law of large numbers,

$$\begin{aligned}
-\frac{1}{n} \log p(x_1, x_2, \ldots, x_n) &= -\frac{1}{n} \sum_{i}^{n} \log p(x_i) \\
&\to -E \log p(X) \\
&= \mathcal{H}.
\end{aligned} \tag{9}$$

$\square$

**Typical Set:** The typical set $\mathcal{A}_\epsilon^{(n)}$ *w.r.t.* $p(x)$ is the set of sequences $(x_1, x_2, \ldots, x_n) \in \mathcal{X}^n$ with the following property:

$$2^{-n(\mathcal{H}+\epsilon)} \le p(x_1, x_2, \ldots, x_n) \le 2^{-n(\mathcal{H}-\epsilon)}, \tag{10}$$

where $\epsilon$ is a constant.

**Properties.** If $(x_1, x_2, \ldots, x_n) \in \mathcal{A}_\epsilon^{(n)}$, we have:

(1). $\mathcal{H}(X) - \epsilon \le -\frac{1}{n} \log p(x_1, x_2, \ldots, x_n) \le \mathcal{H}(X) + \epsilon$, which is determined by the definition of the typical set.

(2). $Pr(\mathcal{A}_\epsilon^{(n)}) > 1 - \epsilon$, for any small number $\epsilon$ together with sufficiently large $n$.

(3). The scale of the typical set $|\mathcal{A}_\epsilon^{(n)}| \leq 2^{n(\mathcal{H}(x)+\epsilon)}$.

(4). $(1-\epsilon)2^{n(\mathcal{H}(x)-\epsilon)} \leq |\mathcal{A}_\epsilon^{(n)}|$ for a sufficiently large $n$.

*Proof.* The proof of property (1) is immediate from the definition of $\mathcal{A}_\epsilon^{(n)}$. The second property follows directly from Theorem 1, since the probability of the sequence $(x_1, x_2, \ldots, x_n) \in \mathcal{A}_\epsilon^{(n)}$ tends to 1 as $n \to \infty$. Thus for any $\delta > 0$, there exists an $n_0$, such that for all $n \geq n_0$, we have

$$Pr\left(\left|-\frac{1}{n}\log p(x_1, x_2, \ldots, x_n) - H(X)\right| < \epsilon\right) > 1 - \delta. \tag{11}$$

We set $\delta = \epsilon$, then obtain the second part of the property. Note that we are using $\epsilon$ for two purposes rather than using both $\epsilon$ and $\delta$. The identification of $\delta = \epsilon$ will conveniently simplify notation later.

To prove property (3), we write

$$\begin{aligned}
1 = \sum_{x \in X^n} p(x) &\geq \sum_{x \in \mathcal{A}_\epsilon^{(n)}} p(x) \\
&\geq \sum_{x \in \mathcal{A}_\epsilon^{(n)}} 2^{-n(H(X+\epsilon))} \\
&= 2^{-n(H(X)+\epsilon)}|\mathcal{A}_\epsilon^{(n)}|,
\end{aligned} \tag{12}$$

where the second inequality follows from Equation 10. Hence $|\mathcal{A}_\epsilon^{(n)}| \leq 2^{n(H(X)+\epsilon)}$.

Finally, for sufficiently large $n$, $Pr(\mathcal{A}_\epsilon^{(n)} > 1 - \epsilon$, so that

$$\begin{aligned}
1 - \epsilon &< Pr(\mathcal{A}_\epsilon^{(n)}) \\
&\leq \sum_{x \in \mathcal{A}_\epsilon^{(n)}} 2^{-n(H(X)-\epsilon)} \\
&= 2^{-n(H(X)-\epsilon)}|\mathcal{A}_\epsilon^{(n)}|,
\end{aligned} \tag{13}$$

hence

$$|\mathcal{A}_\epsilon^{(n)}| \geq (1-\epsilon)2^{n(H(X)-\epsilon)}. \tag{14}$$

This completes the proof of the properties of $\mathcal{A}_\epsilon^{(n)}$ $\qquad\square$

**Lemma 1.** The adversarial example $X$ belongs to the non-typical set.

*Proof.* We define the entropy of normal noise $Z$ in the absence of adversarial interference as $\mathcal{H}(Z)$. Since $Z$ belongs to the typical set, we have

$$\mathcal{H}(Z) = -\frac{1}{n}\log p(z_1, z_2, \ldots, z_n). \tag{15}$$

In the case of adversarial interference, we have

$$p(Z_{adv}) = p(Z|G) = p(z_1|g_1, z_2|g_2, \ldots, z_n|g_n). \tag{16}$$

We further formalize the entropy of $Z_{adv}$ as $\mathcal{H}(Z|G)$. Therefore, the error between two different entropy $\mathcal{H}(Z)$ and $\mathcal{H}(Z|G)$ are shown as follows:

$$\begin{aligned}
\Delta\mathcal{H} &= \mathcal{H}(Z) - \mathcal{H}(Z|G) \\
&= -\sum_z p(z)\log p(z) - \left(-\sum_{z,g} p(z,g)\log p(z,g)\right) \\
&= -\sum_{z,g} p(z,g)\log p(z) + \sum_{z,g} p(z,g)\log p(z,g) \\
&= \sum_{z,g} p(z,g)\log \frac{p(z|g)}{p(z)} \\
&= \sum_{z,g} p(z,g)\log \frac{p(z,g)}{p(z)p(g)} \\
&= \mathcal{I}(Z;G),
\end{aligned} \tag{17}$$

where $\mathcal{I}(Z;G)$ is the mutual information between $Z$ and $G$. From PGD attack Madry et al. (2018), we know that gradient information $G$ is closely related to $Z$. Thus, the value of $\mathcal{I}(Z;G)$ should be greater than zero, leading to:

$$\Delta\mathcal{H} = \mathcal{H}(Z) - \mathcal{H}(Z|G) = \mathcal{I}(Z;G) > 0. \tag{18}$$

That is, $\mathcal{H}(Z) \neq \mathcal{H}(Z|G)$. According to the definition of the AEP, the noise variable $Z_{adv}$ does not satisfy the AEP under adversarial interference. Therefore, the adversarial noise $Z_{adv}$ belongs to the non-typical noise, and the adversarial example $X$ belongs to the non-typical set $\qquad\square$

