# OpenReview forum: "Revisiting Adversarial Examples from the Perspective of Asymptotic Equipartition Property"
_ICLR.cc/2025/Conference — Submitted to ICLR 2025_

### Official Review · Reviewer_FkSG · 2024-11-04

**Soundness:** 1
**Presentation:** 1
**Contribution:** 1
**Rating:** 3
**Confidence:** 3

**Summary:**

This paper introduces the notion of the Asymptotic Equipartition Property (AEP) and then argues that this partitions the input space into a "typical" and "non-typical" set. The paper states that natural examples fall into the typical set, while adversarial examples fall into the non-typical set. The paper argues that adversarial vulnerability arises from

**Strengths:**

The paper addresses an important topic—characterizing the ways in which adversarial examples can be distinguished from natural examples. This remains an open topic and would have important implications if resolved, such as the ability to construct robust classifiers, and further understanding of the nature of adversarial examples.

**Weaknesses:**

This paper is not very clearly written and therefore difficult to evaluate. I think even if all technical issues are addressed, the paper would need to re-written to be reproducible, which, in my opinion, is necessary prior to being publication-ready.
* The notion of "natural" samples and "adversarial" samples is not formally defined, despite being used formally (such as in Lemma 1 & 2). Related, Lemmas 1 & 2 are not stated formally.
* The definitions of Non-Typical I and Non-Typical II are not given.
* The definition of Typical perturbations is not clear, see Questions section.
* It's not precise how the methods were implemented, how exactly is adversarial training performed and how is adversarial accuracy evaluated? (See questions)
* In Table 1, it's not clear what the rows represent—on the top, you have listed three different datasets, and on the bottom, four different models. Which model is used for which dataset? It seems as if the first row of the datasets & of the models is copied.
* Figure 5 doesn't specify which dataset is being used (the numbers seem to indicate it is CIFAR-10, by checking with Table 2, but please confirm this)


The theory in this paper is not rigorous. The main novel theoretical contributions of this paper are Lemma 1 and Lemma 2.
* As discussed above, neither lemmas are stated formally and so it is unclear what is being stated.
* There is no proof of Lemma 1. (In the appendix, there is a proof that is labeled "Lemma 1" but this actually refers to Lemma 2 from the main paper).
* The proof of Lemma 2 (labeled "Lemma 1" in the appendix) does not seem not rigorous: specifically the argument that $I(Z;G)>0$ does not seem to be supported by any formal proof (please see Questions section).


Experimental & Implementation details
* While it is impossible for me to verify, the claimed robustness of the AEP-AT trained models in Table 2 raise a red flag for me to the point where I believe they might indicate a possible bug in the code that generates adversarial attacks—for example, the authors are claiming that the adversarial accuracy. It is somewhat difficult for me to believe that the method proposed by the authors—adding a grid pattern  to adversarial examples during training (as in Figure 4)—would increase adversarial performance by nearly 30 points. In order to verify that the models are actually as robust as the authors claim, the authors should implement adaptive attacks such as AutoAttack. If this is not a bug and holds up under other attack strategies, this is an impressive result. (Nonetheless, my claims about the writing above still stand.)

**Questions:**

On line 811, what is meant by "From PGD attack [Madry et al, 2018] we know that gradient information $G$ is closely related to $Z$." Do you have a reference to what specifically you are referencing from Madry et al., 2018? Is this referring to any formal claim, or just that one can construct an adversarial example by taking gradient steps?

On line 232, you say that for typical set examples, you add AEP-compliant noise, and then give two examples as possible noise distributions, but what specific distribution(s) do you use for your experiments?

One of your primary assumptions is that adversarial examples intervene on the noise component of the data. However, this is the opposite claim of what Ilyas et al., 2019 claims. Can you reconcile your assumption with their claims?

On line 320 you're defining your adversarial accuracy in terms of a PGD attack, but with what perturbation budget? Also, what type of adversarial examples are you using to evaluate the model (e.g., standard adversarial examples or plus {gaussian noise, uniform noise, NT-I, NT-2, etc}?

References:

Andrew Ilyas, Shibani Santurkar, Dimitris Tsipras, Logan Engstrom, Brandon Tran, and Aleksander
Madry. Adversarial examples are not bugs, they are features. In NeurIPS, pp. 125–136, 2019.

---

### Official Review · Reviewer_4bmP · 2024-11-04

**Soundness:** 2
**Presentation:** 2
**Contribution:** 2
**Rating:** 6
**Confidence:** 4

**Summary:**

The paper offers a perspective on adversarial examples by partitioning the input space into typical and non-typical sets. It defines the typical set as the original data samples, while adversarial examples are categorized as non-typical. The authors use this framework to explain various phenomena, including the trade-offs involved in training robust classifiers and the concept of transferability.

**Strengths:**

Comprehensive Conceptualization: The paper presents the concept of adversarial examples from a wide range of research areas within the field, providing a rich and varied perspective.

Engaging Experiments: The authors conduct intriguing experiments related to adversarial training, which enhance the understanding of the proposed concepts and their implications.

**Weaknesses:**

Lack of Comparative Analysis: The paper provides naive empirical results without adequately comparing the proposed adversarial training method to current state-of-the-art approaches (as shown in Table 2 and Figure 5). Additionally, there is insufficient comparison between the results of this method and standard adversarial training (which does not introduce extra noise), as well as a lack of discussion on the balance between clean and adversarial training sets in regular AT (also in Figure 5).

Insufficient Theoretical Discussion: The paper lacks a discussion of existing theories that address the notion of adversarial examples being off the data manifold. There is little exploration of how the concept of non-typical examples relates to the similarity between off-manifold examples and those deemed non-typical.

**Questions:**

1. If random noise is added to training images and still included in the typical set, why is the adversarial (non-typical) set considered larger?

2. How does the AEP-AT trained classifiers perform compared to other existing robust classifiers?

3. What do you perceive as the key differences between off-manifold and non-typical subspaces?

4. Does AEP-AT help reduce overfitting, considering that the generated adversarial examples are more "universal"?

5. Regarding transferability, does this imply that only noise independent of the gradient should be transferable?

---

### Official Review · Reviewer_83o9 · 2024-11-09

**Soundness:** 2
**Presentation:** 3
**Contribution:** 2
**Rating:** 3
**Confidence:** 3

**Summary:**

This paper investigates adversarial perturbations using asymptotic equipartition property (AEP). The authors utilize theoretical tools based on AEP and some related numerical experiments to explain some key different observations in adversarial training, compared with training on clean input, including large model / dataset requirement, adversarial overfitting, clean accuracy / robust accuracy trade-offs and attack transferability.

**Strengths:**

1. It is interesting to investigate the properties of adversarial examples from a probabilistic aspect.

2. The manuscript is relatively well-written and easy to follow, the major message is clear: inputs perturbed by the random noise are typical while the ones perturbed by the adversarial perturbations are non-typical.

**Weaknesses:**

I have the following concerns about this manuscript.

1. My major concern is that the theoretical analyses or explanations in this paper lack some quantitative results or conclusions. From my point of view, normal training (i..e, training on clean inputs) can be considered as a special form of adversarial training when the size of the adversarial budget is $0$. Therefore, any theoretical conclusions concerning the properties of the adversarial perturbations should depend on the size of the adversarial budget. In the formulation of this paper, I can say that the ``adversarial perturbation'' can also be typical when the maximum allowed magnitude is very small. However, the author's analyses and theoretical results do not reflect this.

2. The motivation of AEP-AT is not very clear, the authors need more words to elaborate this part. In addition, the authors claim optimal trade-offs of AEP-AT in line 443, a ROC curve is expected. The advantages of AEP-AT are unclear either.

3. AutoAttack is the strongest adversarial perturbation so far, when evaluating the robustness and reporting robust accuracy in the table, so it is better to evaluate robustness by AutoAttack instead of $10$-step PGD.

4. More empirical simulations are expected for Section 4.3 and 4.4, since the conclusions are already included in existing works.

5. The observations and the conclusions in this paper are already known in this community, making it different to point out the contributions of this paper.

**Questions:**

The questions are pointed out in the "weakness" part above, please take a look and try to address them in the rebuttal. The current version of the manuscript is not ready for publication.

---

### Official Review · Reviewer_Fkq4 · 2024-11-09

**Soundness:** 1
**Presentation:** 2
**Contribution:** 2
**Rating:** 3
**Confidence:** 3

**Summary:**

The paper introduces the concept of asymptotic equipartition property (a direct consequence of the weak law of large numbers) to divide the whole data space into two parts: typical set and non-typical set. Then the paper attempts to show that the normal samples (the raw data perturbed by normal noise) correspond to the typical set and the adversarial samples  (the raw data perturbed by adversarial noise)  correspond to non-typical set. Experiments support their idea and further enhance the understanding of adversarial examples and clarify their counterintuitive phenomena.

**Strengths:**

The basic idea of using the results of the weak law of large numbers to study the behavior of the normal/adversary noises is interesting. The findings based on this idea help us better understand adversarial examples and shed some light on counterintuitive phenomena such as adversarial transferability, the trade-off between robustness and accuracy, and robust overfitting.

**Weaknesses:**

1. My main concern is that the existing basic theoretical foundations of the paper are relatively weak (both Theorems 1 and 2 are known results) and are not clearly explained. I'm trying to understand the theoretical results (especially the proof of Lemma 2), but the vague expression (including the definition of notations, the introduction of background knowledge, and the mathematical analysis and discussion of results) made it difficult for me to understand these results theoretically. See Questions for more details.

2. The paper is not well-written, many concepts and symbols are not defined, theorems and lemmas are not stated rigorously, and many notations are used inconsistently. I would suggest the authors go through the paper and polish the languages and statements.

**Questions:**

1. Theorem 1, the most important concept of the paper, is an informal statement. What is "X" here? Is the convergence based on probability? and the entropy rate is not introduced. In addition, throughout the paper, the notations $H$ and $\mathcal{H}$, $Pr$ and $p$ have been used interchangeably.  Further, is $log$ means $\log_2$? If it is, it would be better to declare it.

2. Page 3, line 148, the paper states that "assuming n $\to \infty$, ......, and all such sequences have the same probability $2^{-nH}$". What does "all such sequences have the same probability" mean?

3. In page 3, when defining "Typical set", $\epsilon$ is simply introduced as a constant, but it is very important, corresponds to the convergence rate in Theorem 1. $\epsilon$ should be discussed carefully here. For example, if the distribution is subGaussian, what is the order of $\epsilon$, and what are the lower and upper bounds of (3).

4. In page 5, why the adversarial noise $Z_{adv}$ is drawn from $P(Z|G)$? how to define $G$?

5. The proof of Lemma 1 should be the proof of Lemma 2, and the proof of Lemma 1 is not given.

6. Page 7, line 346, there is "Section ??".

---

### Meta-Review · Area_Chair_tn6q · 2024-12-17

**Metareview:**

Based on the reviews, I conclude that the paper cannot be accepted for publication in its current form. Of the four reviews, three recommend rejection. The reviewers raised significant concerns regarding key aspects of the work, including the lack of comparative analysis, insufficient theoretical depth, and weak empirical evaluation.

**Additional Comments On Reviewer Discussion:**

Since no rebuttal was provided by the authors, there was no opportunity for discussion or for the authors to address the reviewers’ concerns. As a result, the decision is based solely on the initial submission and the reviewers' feedback without any changes or clarifications from the authors.

---

### Decision · Program_Chairs · 2025-01-22

Reject